# RNNs implicitly implement tensor-product representations

**R. Thomas McCoy,[1] Tal Linzen,[1] Ewan Dunbar,[2] & Paul Smolensky[3,1]**
[1]Department of Cognitive Science, Johns Hopkins University
[2]Laboratoire de Linguistique Formelle, CNRS - Université Paris Diderot - Sorbonne Paris Cité
[3]Microsoft Research AI, Redmond, WA USA
`tom.mccoy@jhu.edu, tal.linzen@jhu.edu,`
`ewan.dunbar@univ-paris-diderot.fr, smolensky@jhu.edu`

## ABSTRACT

Recurrent neural networks (RNNs) can learn continuous vector representations of symbolic structures such as sequences and sentences; these representations often exhibit linear regularities (analogies). Such regularities motivate our hypothesis that RNNs that show such regularities implicitly compile symbolic structures into tensor product representations (TPRs; Smolensky, 1990), which additively combine tensor products of vectors representing roles (e.g., sequence positions) and vectors representing fillers (e.g., particular words). To test this hypothesis, we introduce Tensor Product Decomposition Networks (TPDNs), which use TPRs to approximate existing vector representations. We demonstrate using synthetic data that TPDNs can successfully approximate linear and tree-based RNN autoencoder representations, suggesting that these representations exhibit interpretable compositional structure; we explore the settings that lead RNNs to induce such structure-sensitive representations. By contrast, further TPDN experiments show that the representations of four models trained to encode naturally-occurring sentences can be largely approximated with a bag of words, with only marginal improvements from more sophisticated structures. We conclude that TPDNs provide a powerful method for interpreting vector representations, and that standard RNNs can induce compositional sequence representations that are remarkably well approximated by TPRs; at the same time, existing training tasks for sentence representation learning may not be sufficient for inducing robust structural representations.

## 1 INTRODUCTION

Compositional symbolic representations are widely held to be necessary for intelligence (Newell, 1980; Fodor & Pylyshyn, 1988), particularly in the domain of language (Montague, 1974). However, neural networks have shown great success in natural language processing despite using continuous vector representations rather than explicit symbolic structures. How can these continuous representations yield such success in a domain traditionally believed to require symbol manipulation?

One possible answer is that neural network representations implicitly encode compositional structure. This hypothesis is supported by the spatial relationships between such vector representations, which have been argued to display geometric regularities that parallel plausible symbolic structures of the elements being represented (Mikolov et al. 2013; see Figure 1).

Analogical relationships such as those in Figure 1 are special cases of linearity properties shared by several methods developed in the 1990s for designing compositional vector embeddings of symbolic structures. The most general of these is **tensor product representations** (TPRs; Smolensky 1990). Symbolic structures are first decomposed into **filler-role bindings**; for example, to represent the sequence $[5, 2, 4]$, the filler 5 may be bound to the role of *first element*, the filler 2 may be bound to the role of *second element*, and so on. Each filler $f_i$ and — crucially — each **role** $r_i$ has a vector embedding; these two vectors are combined using their tensor product $f_i \otimes r_i$, and these tensor products are summed to produce the representation of the sequence: $\sum f_i \otimes r_i$. This linear combination can predict the linear relations between sequence representations illustrated in Figure 1.

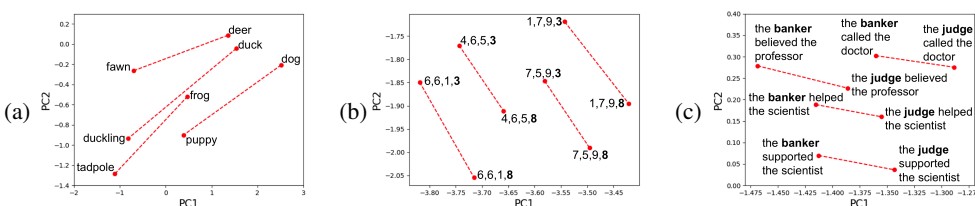

Figure 1: Plots of the first two principal components of (a) word embeddings (Pennington et al., 2014), (b) digit-sequence embeddings learned by an autoencoder (Section 2), and (c) sentences (InferSent: Conneau et al. 2017). All demonstrate systematicity in the learned vector spaces.

In this article, we test the hypothesis that vector representations of sequences can be approximated as a sum of filler-role bindings, as in TPRs. We introduce the Tensor Product Decomposition Network (TPDN) which takes a set of continuous vector representations to be analyzed and learns filler and role embeddings that best predict those vectors, given a particular hypothesis for the relevant set of roles (e.g., sequence indexes or structural positions in a parse tree).

To derive structure-sensitive representations, in Section 2 we look at a task driven by structure, not content: autoencoding of sequences of meaningless symbols, denoted by digits. The focus here is on sequential structure, although we also devise a version of the task that uses tree structure. For the representations learned by these autoencoders, TPDNs find excellent approximations that are TPRs.

In Section 3, we turn to sentence-embedding models from the contemporary literature. It is an open question how structure-sensitive these representations are; to the degree that they are structure-sensitive, our hypothesis is that they can be approximated by TPRs. Here, TPDNs find less accurate approximations, but they also show that a TPR equivalent to a bag-of-words already provides a reasonable approximation; these results suggest that these sentence representations are not robustly structure-sensitive. We therefore return to synthetic data in Section 4, exploring which architectures and training tasks are likely to lead RNNs to induce structure-sensitive representations.

To summarize the contributions of this work, TPDNs provide a powerful method for interpreting vector representations, shedding light on hard-to-understand neural architectures. We show that standard RNNs can induce compositional representations that are remarkably well approximated by TPRs and that the nature of these representations depends, in intrepretable ways, on the architecture and training task. Combined with our finding that standard sentence encoders do not seem to learn robust representations of structure, these findings suggest that more structured architectures or more structure-dependent training tasks could improve the compositional capabilities of existing models.

## 1.1 THE TENSOR PRODUCT DECOMPOSITION NETWORK

The Tensor Product Decomposition Network (TPDN), depicted in Figure 2c, learns a TPR that best approximates an existing set of vector encodings. While TPDNs can be applied to any structured space, including embeddings of images or words, this work focuses on applying TPDNs to sequences. The model is given a hypothesized role scheme and the dimensionalities of the filler and role embeddings. The elements of each sequence are assumed to be the fillers in that sequence's representation; for example, if the hypothesized roles are indexes counting from the end of the sequence, then the hypothesized filler-role pairs for $[5, 2, 4]$ would be (4:*last*, 2:*second-to-last*, 5:*third-to-last*).

The model then learns embeddings for these fillers and roles that minimize the distance between the TPRs generated from these embeddings and the existing encodings of the sequences. Before the comparison is performed, the tensor product (which is a matrix) is flattened into a vector, and a linear transformation $M$ is applied (see Appendix B for an ablation study showing that this transformation, which was not a part of the original TPR proposal, is necessary). The overall function computed by the architecture is thus $M(\textit{flatten}(\sum_i r_i \otimes f_i))$.

PyTorch code for the TPDN model is available on GitHub,[1] along with an interactive demo.[2]

---

[1] https://github.com/tommccoy1/tpdn
[2] https://tommccoy1.github.io/tpdn/tpr_demo.html

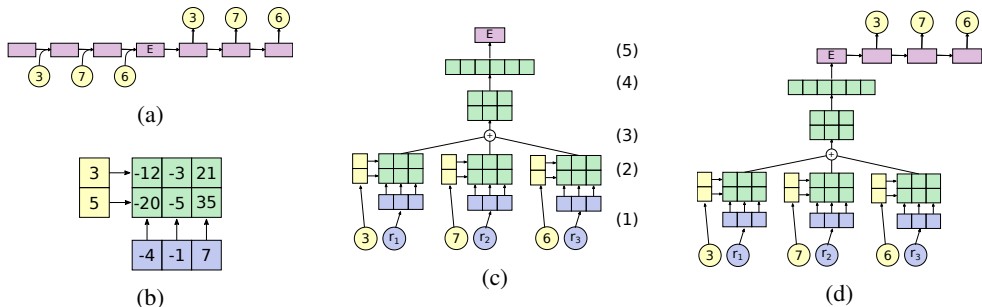

Figure 2: **(a)** A unidirectional sequence-to-sequence autoencoder. **(b)** The tensor product operation. **(c)** A TPDN trained to approximate the encoding $E$ from the autoencoder: (1) The fillers and roles are embedded. (2) The fillers and roles are bound together using the tensor product. (3) The tensor products are summed. (4) The sum is flattened into a vector by concatenating the rows. (5) A linear transformation is applied to get the final encoding. **(d)** The architecture for evaluation: using the original autoencoder's decoder with the trained TPDN as the encoder.

## 2 APPROXIMATING RNN AUTOENCODER REPRESENTATIONS

To establish the effectiveness of the TPDN at uncovering the structural representations used by RNNs, we first apply the TPDN to sequence-to-sequence networks trained on an autoencoding objective: they are expected to encode a sequence of digits and then decode that encoding to reproduce the same sequence (Figure 2a). In addition to testing the TPDN, this experiment also addresses a scientific question: do different architectures (specifically, unidirectional, bidirectional, and tree-based sequence-to-sequence models) induce different representations?

### 2.1 EXPERIMENTAL SETUP

**Digit sequences:** The sequences consisted of the digits from 0 to 9. We randomly generated 50,000 unique sequences with lengths ranging from 1 to 6 inclusive and averaging 5.2; these sequences were divided into 40,000 training sequences, 5,000 development sequences, and 5,000 test sequences.

**Architectures:** For all sequence-to-sequence networks, we used gated recurrent units (GRUs, Cho et al. (2014)) as the recurrent units. We considered three encoder-decoder architectures: unidirectional, bidirectional, and tree-based.[3] The unidirectional encoders and decoders follow the setup of Sutskever et al. (2014): the encoder is fed the input elements one at a time, left to right, updating its hidden state after each element. The decoder then produces the output sequence using the final hidden state of the encoder as its input. The bidirectional encoder combines left-to-right and right-to-left unidirectional encoders (Schuster & Paliwal, 1997); for symmetry, we also create a bidirectional decoder, which has both a left-to-right and a right-to-left unidirectional decoder whose hidden states are concatenated to form bidirectional hidden states from which output predictions are made. Our final topology is tree-based RNNs (Pollack, 1990; Socher et al., 2010), specifically the Tree-GRU encoder of Chen et al. (2017) and the tree decoder of Chen et al. (2018). These architectures require a tree structure as part of their input; we generated a tree for each sequence using a deterministic algorithm that groups digits based on their values (see Appendix C). To control for initialization effects, we trained five instances of each architecture with different random initializations.

**Role schemes:** We consider 6 possible methods that networks might use to represent the roles of specific digits within a sequence; see Figure 3a for examples of these role schemes.

1. **Left-to-right:** Each digit's role is its index in the sequence, counting from left to right.
2. **Right-to-left:** Each digit's role is its index in the sequence, counting from right to left.
3. **Bidirectional:** Each digit's role is an ordered pair containing its left-to-right index and its right-to-left index (compare human representations of spelling, Fischer-Baum et al. 2010).
4. **Wickelroles:** Each digit's role is the digit before it and the digit after it (Wickelgren, 1969).

---

[3]For this experiment, the encoder and decoder always matched in type.

5. **Tree positions:** Each digit's role is its position in a tree, such as RRL (left child of right child of right child of root). The tree structures are given by the algorithm in Appendix C.

6. **Bag-of-words:** All digits have the same role. We call this a *bag-of-words* because it represents which digits ("words") are present and in what quantities, but ignores their positions.

**Hypothesis:** We hypothesize that RNN autoencoders will learn to use role representations that parallel their architectures: left-to-right roles for a unidirectional network, bidirectional roles for a bidirectional network, and tree-position roles for a tree-based network.

**Evaluation:** We evaluate how well a given sequence-to-sequence network can be approximated by a TPR with a particular role scheme as follows. First, we train a TPDN with the role scheme in question (Section 1.1). Then, we take the original encoder/decoder network and substitute the fitted TPDN for its encoder (Figure 2d). We do not conduct any additional training upon this hybrid network; the decoder retains exactly the weights it learned in association with the original encoder, while the TPDN retains exactly the weights it learned for approximating the original encoder (including the weights on the final linear layer). We then compute the accuracy of the resulting hybrid network; we call this metric the **substitution accuracy**. High substitution accuracy indicates that the TPDN has approximated the encoder well enough for the decoder to handle the resulting vectors.

## 2.2 RESULTS

**Performance of seq2seq networks:** The unidirectional and tree-based architectures both performed the training task nearly perfectly, with accuracies of 0.999 and 0.989 (averaged across five runs). Accuracy was lower (0.834) for the bidirectional architecture; this might mean that the hidden size of 60 becomes too small when divided into two 30-dimensional halves, one half for each direction.

**Quality of TPDN approximation:** For each of the six role schemes, we fitted a TPDN to the vectors generated by the trained encoder, and evaluated it using substitution accuracy (Section 2.1). The results, in Figure 3c, show that different architectures do use different representations to solve the task. The tree-based autoencoder can be well-approximated using tree-position roles but not using any of the other role schemes. By contrast, the unidirectional architecture is approximated very closely (with a substitution accuracy of over 0.99 averaged across five runs) by bidirectional roles. Left-to-right roles are also fairly successful (accuracy = 0.87), and right-to-left roles are decidedly unsuccessful (accuracy = 0.11). This asymmetry suggests that the unidirectional network

| | 3 | 1 | 1 | 6 | 5 | 2 | 3 | 1 | 9 | 7 |
|---|---|---|---|---|---|---|---|---|---|---|
| Left-to-right | 0 | 1 | 2 | 3 | 0 | 1 | 2 | 3 | 4 | 5 |
| Right-to-left | 3 | 2 | 1 | 0 | 5 | 4 | 3 | 2 | 1 | 0 |
| Bidirectional | (0, 3) | (1, 2) | (2, 1) | (3, 0) | (0, 5) | (1, 4) | (2, 3) | (3, 2) | (4, 1) | (5, 0) |
| Wickelroles | #_1 | 3_1 | 1_6 | 1_# | #_2 | 5_3 | 2_1 | 3_9 | 1_7 | 9_# |
| Tree | L | RLL | RLR | RR | LL | LRLL | LRLR | LRRL | LRRR | R |
| Bag of words | $r_0$ | $r_0$ | $r_0$ | $r_0$ | $r_0$ | $r_0$ | $r_0$ | $r_0$ | $r_0$ | $r_0$ |

(a)

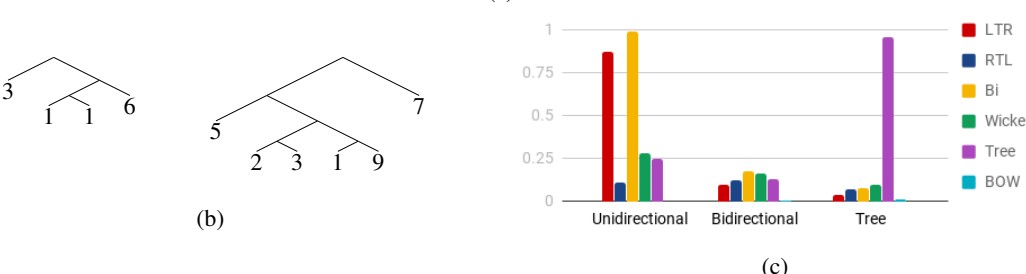

(b)

(c)

Figure 3: (a) The filler-role bindings assigned by the six role schemes to two sequences, 3116 and 523197. Roles not shown are assigned the null filler. (b) The trees used to assign tree roles to these sequences. (c) Substitution accuracy for three architectures at the autoencoding task with six role schemes. Each bar represents an average across five random initializations.

uses *mildly bidirectional* roles: while it is best approximated by bidirectional roles, it strongly favors one direction over the other. Though the model uses bidirectional roles, then, roles with the same left-to-right position (e.g. (2,3), (2,4), and (2,5)) can be collapsed without much loss of accuracy.

Finally, the bidirectional architecture is not approximated well by any of the role schemes we investigated. It may be implementing a role scheme we did not consider, or a structure-encoding scheme other than TPR. Alternately, it might simply not have adopted any robust method for representing sequence structure; this could explain why its accuracy on the training task was relatively low (0.83).

## 3 Encodings of naturally-occurring sentences

Will the TPDN's success with digit-sequence autoencoders extend to models trained on naturally occurring data? We explore this question using sentence representations from four models: InferSent (Conneau et al., 2017), a BiLSTM trained on the Stanford Natural Language Inference (SNLI) corpus (Bowman et al., 2015); Skip-thought (Kiros et al., 2015), an LSTM trained to predict the sentence before or after a given sentence; the Stanford sentiment model (SST) (Socher et al., 2013), a tree-based recursive neural tensor network trained to predict movie review sentiment; and SPINN (Bowman et al., 2016), a tree-based RNN trained on SNLI. More model details are in Appendix E.

### 3.1 TPDN approximation

We now fit TPDNs to these four sentence encoding models. We experiment with all of the role schemes used in Section 2 except for Wickelroles; for sentence representations, the vocabulary size $|V|$ is so large that the Wickelrole scheme, which requires $|V|^2$ distinct roles, becomes intractable.

Preliminary experiments showed that the TPDN performed poorly when learning the filler embeddings from scratch, so we used pretrained word embeddings; for each model, we use the word embeddings used by that model. We fine-tuned the embeddings with a linear transformation on top of the word embedding layer (though the embeddings themselves remain fixed). Thus, what the model has to learn are: the role embeddings, the linear transformation to apply to the fixed filler embeddings, and the final linear transformation applied to the sum of the filler/role bindings.

We train TPDNs on the sentence embeddings that each model generates for all SNLI premise sentences (Bowman et al., 2015). For other training details see Appendix E. Table 1a shows the mean squared errors (MSEs) for various role schemes. In general, the MSEs show only small differences between role schemes, except that tree-position roles do noticeably outperform other role schemes for SST. Notably, bag-of-words roles perform nearly as well as the other role schemes, in stark contrast to the poor performance of bag-of-words roles in Section 2. MSE is useful for comparing models but is less useful for assessing absolute performance since the exact value of this error is not very interpretable. In the next section, we use downstream tasks for a more interpretable evaluation.

### 3.2 Performance on downstream tasks

**Tasks:** We assess how the tensor product approximations compare to the models they approximate at four tasks that are widely accepted for evaluating sentence embeddings: (1) Stanford Sentiment Treebank (SST), rating the sentiment of movie reviews (Socher et al., 2013); (2) Microsoft Research

|     |              | LTR  | RTL  | Bi   | Tree     | BOW  |
| --- | ------------ | ---- | ---- | ---- | -------- | ---- |
| (a) | InferSent    | 0.17 | 0.18 | 0.17 | **0.16** | 0.19 |
|     | Skip-thought | 0.45 | 0.46 | 0.47 | **0.42** | 0.45 |
|     | SST          | 0.24 | 0.26 | 0.26 | **0.17** | 0.27 |
|     | SPINN        | 0.22 | 0.23 | 0.21 | **0.18** | 0.25 |

|     |              | LTR  | RTL  | Bi       | Tree | BOW  |
| --- | ------------ | ---- | ---- | -------- | ---- | ---- |
| (b) | InferSent    | 0.35 | 0.34 | **0.29** | 0.35 | 0.40 |
|     | Skip-thought | 0.34 | 0.37 | **0.24** | 0.34 | 0.51 |
|     | SST          | 0.27 | 0.32 | **0.25** | 0.26 | 0.34 |
|     | SPINN        | 0.49 | 0.53 | **0.44** | 0.49 | 0.56 |

Table 1: (a) MSEs of TPDN approximations of sentence encodings (normalized by dividing by the MSE from training the TPDN on random vectors, to allow comparisons across models). (b) Performance of the sentence encoding models on our role-diagnostic analogies. Numbers indicate Euclidean distances (normalized by dividing by the average distance between vectors in the analogy set). Each column contains the average over all analogies diagnostic of the role heading that column.

| | Model | LTR | RTL | Bi | Tree | BOW | | Model | LTR | RTL | Bi | Tree | BOW |
|---|---|---|---|---|---|---|---|---|---|---|---|---|---|
| (a) | InferSent | **0.79** | **0.79** | 0.78 | 0.78 | 0.77 | (b) | InferSent | **0.77** | 0.74 | **0.77** | **0.77** | 0.71 |
| | Skip-thought | 0.53 | 0.52 | 0.46 | 0.50 | **0.58** | | Skip-thought | **0.37** | **0.37** | 0.36 | 0.36 | **0.37** |
| | SST | **0.83** | 0.82 | 0.82 | 0.82 | 0.81 | | SST | 0.48 | 0.51 | 0.49 | **0.67** | 0.49 |
| | SPINN | 0.73 | 0.75 | 0.75 | **0.76** | 0.74 | | SPINN | 0.72 | 0.72 | 0.73 | **0.76** | 0.58 |

Table 2: The proportion of test examples on which a classifier trained on sentence encodings gave the same predictions for the original encodings and for their TPDN approximations. (a) shows the average of these proportions across SST, MRPC, and STS-B, while (b) shows only SNLI. (For including STS-B in (a), we linearly shift its values to be in the same range as the other tasks' results).

Paraphrase Corpus (MRPC), classifying whether two sentences paraphrase each other (Dolan et al., 2004); (3) Semantic Textual Similarity Benchmark (STS-B), labeling how similar two sentences are (Cer et al., 2017); and (4) Stanford Natural Language Inference (SNLI), determining if one sentence entails a second sentence, contradicts the second sentence, or neither (Bowman et al., 2015).

**Evaluation:** We use SentEval (Conneau & Kiela, 2018) to train a classifier for each task on the original encodings produced by the sentence encoding model. We freeze this classifier and use it to classify the vectors generated by the TPDN. We then measure what proportion of the classifier's predictions for the approximation match its predictions for the original sentence encodings.[4]

**Results:** For all tasks besides SNLI, we found no marked difference between bag-of-words roles and other role schemes (Table 2a). For SNLI, we did see instances where other role schemes outperformed bag-of-words (Table 2b). Within the SNLI results, both tree-based models (SST and SPINN) are best approximated with tree-based roles. InferSent is better approximated with structural roles than with bag-of-words roles, but all structural role schemes perform similarly. Finally, Skip-thought cannot be approximated well with any role scheme we considered. It is unclear why Skip-thought has lower results than the other models. Overall, even for SNLI, bag-of-words roles provide a fairly good approximation, with structured roles yielding rather modest improvements.

Based on these results, we hypothesize that these models' representations can be characterized as a bag-of-words representation plus some incomplete structural information that is not always encoded. This explanation is consistent with the fact that bag-of-words roles yield a strong but imperfect approximation for the sentence embedding models. However, this is simply a conjecture; it is possible that these models do use a robust, systematic structural representation that either involves a role scheme we did not test or that cannot be characterized as a tensor product representation at all.

### 3.3 ANALOGIES

We now complement the TPDN tests with sentence analogies. By comparing pairs of minimally different sentences, analogies might illuminate representational details that are difficult to discern in individual sentences. We construct sentence-based analogies that should hold only under certain role schemes, such as the following analogy (expressed as an equation as in Mikolov et al. 2013):

$$\textbf{I see now} - \textbf{I see} = \textbf{you know now} - \textbf{you know} \tag{1}$$

A left-to-right role scheme makes (1) equivalent to (2) ($f{:}r$ denotes the binding of filler $f$ to role $r$):

$$(\text{I:0} + \text{see:1} + \text{now:2}) - (\text{I:0} + \text{see:1}) = (\text{you:0} + \text{know:1} + \text{now:2}) - (\text{you:0} + \text{know:1}) \tag{2}$$

In (2), both sides reduce to now:2, so (1) holds for representations using left-to-right roles. However, if (2) instead used right-to-left roles, it would not reduce in any clean way, so (1) would not hold. We construct a dataset of such role-diagnostic analogies, where each analogy should only hold for certain role schemes. For example, (1) works for left-to-right roles or bag-of-words roles, but not the other role schemes. The analogies use a vocabulary based on Ettinger et al. (2018) to ensure plausibility of the constructed sentences. For each analogy, we create 4 equations, one isolating

---

[4]We also train SentEval classifiers on top of the TPDN instead of the original model; see Appendix F for these results. In general, for all models besides Skip-thought, the TPDN approximations perform nearly as well as the original models, and in some cases the approximations even outperform the originals.

each of the four terms (e.g. **I see = I see now – you know now + you know**). We then compute the Euclidean distance between the two sides of each equation using each model's encodings.

The results are in Table 1b. InferSent, Skip-thought, and SPINN all show results most consistent with bidirectional roles, while SST shows results most consistent with tree-based or bidirectional roles. The bag-of-words column shows poor performance by all models, indicating that in controlled enough settings these models can be shown to have some more structured behavior even though evaluation on examples from applied tasks does not clearly bring out that structure. These analogies thus provide independent evidence for our conclusions from the TPDN analysis: these models have a weak notion of structure, but that structure is largely drowned out by the non-structure-sensitive, bag-of-words aspects of their representations. However, the other possible explanations mentioned above−namely, the possibilities that the models use alternate role schemes that we did not test or that they use some structural encoding other than tensor product representation−still remain.

## 4    WHEN DO RNNS LEARN COMPOSITIONAL REPRESENTATIONS?

The previous section suggested that all sentence models surveyed did not robustly encode structure and could even be approximated fairly well with a bag of words. Motivated by this finding, we now investigate how aspects of training can encourage or discourage compositionality in learned representations. To increase interpretability, we return to the setting (from Section 2) of operating over digit sequences. We investigate two aspects of training: the architecture and the training task.

**Teasing apart the contribution of the encoder and decoder:**   In Section 2, we investigated autoencoders whose encoder and decoder had the same topology (unidirectional, bidirectional, or tree-based). To test how each of the two components contributes to the learned representation, we now expand the investigation to include networks where the encoder and decoder differ. We crossed all three encoder types with all three decoder types (nine architectures in total). The results are in Table 7 in Appendix D. The decoder largely dictates what roles are learned: models with unidirectional decoders prefer mildly bidirectional roles, models with bidirectional decoders fail to be well-approximated by any role scheme, and models with tree-based decoders are best approximated by tree-based roles. However, the encoder still has some effect: in the tree/uni and tree/bi models, the tree-position roles perform better than they do for the other models with the same decoders. Though work on novel architectures often focuses on the encoder, this finding suggests that focusing on the decoder may be more fruitful for getting neural networks to learn specific types of representations.

**The contribution of the training task:**   We next explore how the training task affects the representations that are learned. We test four tasks, illustrated in Table 3a: **autoencoding** (returning the input sequence unchanged), **reversal** (reversing the input), **sorting** (returning the input digits in ascending order), and **interleaving** (alternating digits from the left and right edges of the input).

Table 3b gives the substitution accuracy for a TPDN trained to approximate a unidirectional encoder that was trained with a unidirectional decoder on each task. Training task noticeably influences the learned representations. First, though the model has learned mildly bidirectional roles favoring the left-to-right direction for autoencoding, for reversal the right-to-left direction is far preferred over left-to-right. For interleaving, the model is approximated best with strongly bidirectional roles: that is, bidirectional roles work nearly perfectly, while neither unidirectional scheme works well. Finally, for sorting, bag-of-words roles work nearly as well as all other schemes, suggesting that the model

| | Input | | | | | LTR | RTL | Bi | Wickel | Tree | BOW |
|---|---|---|---|---|---|---|---|---|---|---|---|
| | Input | 3,4,0 | 4,3,6,5,1,3 | | | | | | | | |
| (a) | Autoencode | 3,4,0 | 4,3,6,5,1,3 | (b) | Autoencode | 0.87 | 0.11 | **0.99** | 0.28 | 0.25 | 0.00 |
| | Reverse | 0,4,3 | 3,1,5,6,3,4 | | Reverse | 0.06 | 0.99 | **1.00** | 0.18 | 0.20 | 0.00 |
| | Sort | 0,3,4 | 1,3,3,4,5,6 | | Sort | 0.90 | 0.90 | **0.92** | 0.88 | 0.89 | 0.89 |
| | Interleave | 3,0,4 | 4,3,3,1,6,5 | | Interleave | 0.27 | 0.18 | **0.99** | 0.63 | 0.36 | 0.00 |

Table 3: (a) Tasks used to test for the effect of task on learned roles (Section 4). (b) Accuracy of the TPDN applied to models trained on these tasks with a unidirectional encoder and decoder. All numbers are averages across five random initializations.

has learned to discard most structural information since sorting does not depend on structure. These experiments suggest that RNNs only learn compositional representations when the task requires them. This result might explain why the sentence embedding models do not seem to robustly encode structure: perhaps the training tasks for these models do not heavily rely on sentence structure (e.g. Parikh et al. (2016) achieved high accuracy on SNLI using a model that ignores word order), such that the models learn to ignore structural information, as was the case with models trained on sorting.

## 5    RELATED WORK

There are several approaches for interpreting neural network representations. One approach is to infer the information encoded in the representations from the system's behavior on examples targeting specific representational components, such as semantics (Pavlick, 2017; Dasgupta et al., 2018; Poliak et al., 2018) or syntax (Linzen et al., 2016). Another approach is based on probing tasks, which assess what information can be easily decoded from a vector representation (Shi et al. 2016; Belinkov et al. 2017; Kádár et al. 2017; Ettinger et al. 2018; compare work in cognitive neuroscience, e.g. Norman et al. 2006). Our method is wider-reaching than the probing task approach, or the Mikolov et al. (2013) analogy approach: instead of decoding a single feature, we attempt to exhaustively decompose the vector space into a linear combination of filler-role bindings.

The TPDN's successful decomposition of sequence representations in our experiments shows that RNNs can sometimes be approximated with no nonlinearities or recurrence. This finding is related to the conclusions of Levy et al. (2018), who argued that LSTMs dynamically compute weighted sums of their inputs; TPRs replace the weights of the sum with the role vectors. Levy et al. (2018) also showed that recurrence is largely unnecessary for practical applications. Vaswani et al. (2017) report very good performance for a sequence model without recurrence; importantly, they find it necessary to incorporate sequence position embeddings, which are similar to the left-to-right roles discussed in Section 2. Methods for interpreting neural networks using more interpretable architectures have been proposed before based on rules and automata (Omlin & Giles, 1996; Weiss et al., 2018).

Our decomposition of vector representations into independent fillers and roles is related to work on separating latent variables using singular value decomposition and other factorizations (Tenenbaum & Freeman, 2000; Anandkumar et al., 2014). For example, in face recognition, eigenfaces (Sirovich & Kirby, 1987; Turk & Pentland, 1991) and TensorFaces (Vasilescu & Terzopoulos, 2002; 2005) use such techniques to disentangle facial features, camera angle, and lighting.

Finally, there is a large body of work on incorporating explicit symbolic representations into neural networks (for a recent review, see Battaglia et al. 2018); indeed, tree-shaped RNNs are an example of this approach. While our work is orthogonal to this line of work, we note that TPRs and other filler-role representations can profitably be used as an explicit component of neural models (Koniusz et al., 2017; Palangi et al., 2018; Huang et al., 2018; Tang et al., 2018; Schlag & Schmidhuber, 2018).

## 6    CONCLUSION

What kind of internal representations could allow simple sequence-to-sequence models to perform the remarkable feats they do, including tasks previously thought to require compositional, symbolic representations (e.g., translation)? Our experiments show that, in heavily structure-sensitive tasks, sequence-to-sequence models learn representations that are extremely well approximated by tensor-product representations (TPRs), distributed embeddings of symbol structures that enable powerful symbolic computation to be performed with neural operations (Smolensky, 2012). We demonstrated this by approximating learned representations via TPRs using the proposed tensor-product decomposition network (TPDN). Variations in architecture and task were shown to induce different types and degrees of structure-sensitivity in representations, with the decoder playing a greater role than the encoder in determining the structure of the learned representation. TPDNs applied to mainstream sentence-embedding models reveal that unstructured bag-of-words models provide a respectable approximation; nonetheless, this experiment also provides evidence for a moderate degree of structure-sensitivity. The presence of structure-sensitivity is corroborated by targeted analogy tests motivated by the linearity of TPRs. A limitation of the current TPDN architecture is that it requires a hypothesis about the representations to be selected in advance. A fruitful future research direction would be to automatically explore hypotheses about the nature of the TPR encoded by a network.

ACKNOWLEDGMENTS

This material is based upon work supported by the National Science Foundation Graduate Research Fellowship Program under Grant No. 1746891 and NSF INSPIRE grant BCS-1344269. This work was also supported by ERC grant ERC-2011-AdG-295810 (BOOTPHON), and ANR grants ANR-10-LABX-0087 (IEC) and ANR-10-IDEX-0001-02 (PSL*), ANR-17-CE28-0009 (GEOMPHON), ANR-11-IDEX-0005 (USPC), and ANR-10-LABX-0083 (EFL). Any opinions, findings, and conclusions or recommendations expressed in this material are those of the authors and do not necessarily reflect the views of the National Science Foundation or the other supporting agencies.

For helpful comments, we are grateful to Colin Wilson, John Hale, Marten van Schijndel, Jan Hůla, the members of the Johns Hopkins Gradient Symbolic Computation research group, and the members of the Deep Learning Group at Microsoft Research, Redmond. Any errors remain our own.

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

## A    LIST OF ACRONYMS AND ABBREVIATIONS

| | |
|---|---|
| Bi | Bidirectional |
| BOW | Bag of words |
| LTR | Left to right |
| MRPC | Microsoft Research Paraphrase Corpus |
| RTL | Right to left |
| SNLI | Stanford Natural Language Inference corpus |
| SST | Stanford Sentiment Treebank |
| STS-B | Semantic Textual Similarity Benchmark |
| TPDN | Tensor product decomposition network |
| TPR | Tensor product representation |
| Uni | Unidirectional |
| Wickel | Wickelroles (see Section 2.1) |

## B    ANALYSIS OF ARCHITECTURE COMPONENTS

Here we analyze how several aspects of the TPDN architecture contribute to our results. For all of the experiements described in this section, we used TPDNs to approximate a sequence-to-sequence network with a unidirectional encoder and unidirectional decoder that was trained to perform the reversal task (Section 4); we chose this network because it was strongly approximated by right-to-left roles, which are relatively simple (but still non-trivial).

### B.1    IS THE FINAL LINEAR LAYER NECESSARY?

One area where our model diverges from traditional tensor product representations is in the presence of the final linear layer (step 5 in Figure 2c). This layer is necessary if one wishes to have freedom to choose the dimensionality of the filler and role embeddings; without it, the dimensionality of the representations that are being approximated must factor exactly into the product of the dimensionality of the filler embeddings and the dimensionality of the role embedding (see Figure 2c). It is natural to wonder whether the only contribution of this layer is in adjusting the dimensionality or whether it serves a broader function. Table 4 shows the results of approximating the reversal sequence-to-sequence network with and without this layer; it indicates that this layer is highly necessary for the successful decomposition of learned representations. (Tables follow all appendix text.)

### B.2    VARYING THE DIMENSIONALITY OF THE FILLER AND ROLE EMBEDDINGS

Two of the parameters that must be provided to the TPDN are the dimensionality of the filler embeddings and the dimensionality of the role embeddings. We explore the effects of these parameters in Figure 4. For the role embeddings, substitution accuracy increases noticeably with each increase in dimensionality until the dimensionality hits 6, where accuracy plateaus. This behavior is likely due to the fact that the reversal seq2seq network is most likely to employ right-to-left roles, which involves 6 possible roles in this setting. A dimensionality of 6 is therefore the minimum embedding size needed to make the role vectors linearly independent; linear independence is an important property for the fidelity of a tensor product representation (Smolensky, 1990). The accuracy also generally increases as filler dimensionality increases, but there is a less clear point where it plateaus for the fillers than for the roles.

### B.3    FILLER-ROLE BINDING OPERATION

The body of the paper focused on using the tensor product ($f_i \otimes r_i$, see Figure 2b) as the operation for binding fillers to roles. There are other conceivable binding operations. Here we test two alternatives, both of which can be viewed as special cases of the tensor product or as related to it: circular convolution, which is used in holographic reduced representations (Plate, 1995), and elementwise product ($f_i \odot r_i$). Both of these are restricted such that roles and fillers must have the same embedding dimension ($N_f = N_r$). We first try setting this dimension to 20, which is what was used as both the role and filler dimension in all tensor product experiments with digit sequences.

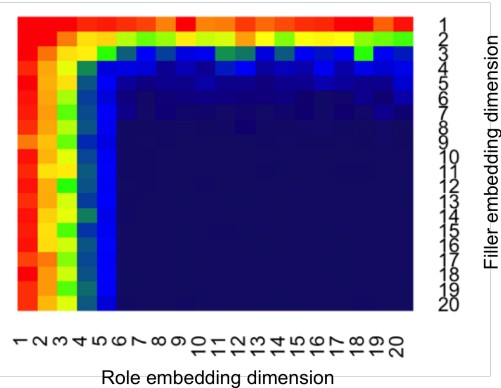

Figure 4: Heatmap of substitution accuracies with various filler and role embedding dimensions. Red indicates accuracy under 1%; dark blue indicates accuracy over 80%. The models whose substitution accuracies are displayed are all TPDNs trained to approximate a sequence-to-sequence model that was trained on the task of reversal.

We found that while these dimensions were effective for the tensor product binding operation, they were not effective for elementwise product and circular convolution (Table 5). When the dimension was increased to 60, however, the elementwise product performed roughly as well as as the tensor product; circular convolution now learned one of the two viable role schemes (right-to-left roles) but failed to learn the equally viable bidirectional role scheme. Thus, our preliminary experiments suggest that these other two binding operations do show promise, but seem to require larger embedding dimensions than tensor products do. At the same time, they still have fewer parameters overall compared to the tensor product because their final linear layers (of dimensionality $N$) are much smaller than those used with a tensor product (of dimensionality $N^2$).

## C  THE DIGIT PARSING ALGORITHM

When inputting digit sequences to our tree-based model, the model requires a predefined tree structure for the digit sequence. We use the following algorithm to generate this tree structure: at each timestep, combine the smallest element of the sequence (other than the last element) with its neighbor immediately to the right, and replace the pair with that neighbor. If there is a tie for the smallest digit, choose the leftmost tied digit.

For example, the following shows step-by-step how the tree for the sequence 523719 would be generated:

- 5 2 3 7 1 9
- 5 2 3 7 [1 9]
- 5 [2 3] 7 [1 9]
- 5 [ [2 3] 7] [1 9]
- [ 5 [ [2 3] 7] [1 9] ]

## D  FULL RESULTS OF SEQUENCE-TO-SEQUENCE EXPERIMENTS

Section 4 summarized the results of our experiments which factorially varied the training task, the encoder and the decoder. Here we report the full results of these experiments in two tables: Table 6 shows the accuracies achieved by the sequence-to-sequence models at the various training tasks, and Table 7 shows the substitution accuracies of TPDNs applied to the trained sequence-to-sequence models for all architectures and tasks.

## E  MODEL AND TRAINING DETAILS

### E.1  SEQUENCE-TO-SEQUENCE MODELS

As much as possible, we standardized parameters across all sequence-to-sequence models that we trained on the digit-sequence tasks.

For all decoders, when computing a new hidden state, the only input to the recurrent unit is the previous hidden state (or parent hidden state, for a tree-based decoder), without using any previous outputs as inputs to the hidden state update. This property is necessary for using a bidirectional decoder, since it would not be possible to generate the output both before and after each bidirectional decoder hidden state.

We also inform the decoder of when to stop decoding; that is, for sequential models, the decoder stops once its output is the length of the sequence, while for tree-based models we tell the model which positions in the tree are leaves. Stopping could alternately be determined by some action of the decoder (e.g., generating an end-of-sequence symbol); for simplicity we chose the strategy outlined above instead.

For all architectures, we used a digit embedding dimensionality of 10 (chosen arbitrarily) and a hidden layer size of 60 (this hidden layer size was chosen because 60 has many integer factors, making it amenable to the dimensionality analyses in Appendix B.2). For the bidirectional architectures, the forward and backward recurrent layers each had a hidden layer size of 30, so that their concatenated hidden layer size was 60. For bidirectional decoders, a linear layer condensed the 60-dimensional encoding into 30 dimensions before it was passed to the forward and backward decoders.

The networks were trained using the Adam optimizer (Kingma & Ba, 2015) with the standard initial learning rate of 0.001. We used negative log likelihood, computed over the softmax probability distributions for each output sequence element, as the loss function. Training proceeded with a batch size of 32, with loss on the held out development set computed after every 1,000 training examples. Training was halted when the loss on the heldout development set had not improved for any of the development loss checkpoints for a full epoch of training (i.e. 40,000 training examples). Once training completed, the parameters from the best-performing checkpoint were reloaded and used for evaluation of the network.

### E.2  TPDNs TRAINED ON DIGIT MODELS

When applying TPDNs to the digit-based sequence-to-sequence models, we always used 20 as both the filler embedding dimension and the role embedding dimension. This decision was based on the experiments in Appendix B.2; we selected filler and role embedding dimensions that were safely above the cutoff needed to lead to successful decomposition.

The TPDNs were trained with the same training regimen as the sequence-to-sequence models, except that, instead of using negative log likelihood as the loss function, for the TPDNs we used mean squared error between the predicted vector representation and the actual vector representation from the original sequence-to-sequence network.

The TPDNs were given the sequences of fillers (i.e. the digits), the roles hypothesized to go with those fillers, the sequence embeddings produced by the RNN, and the dimensionalities of the filler embeddings, role embeddings, and final linear transformation. The parameters that were updated by training were the specific values for the filler embeddings, the role embeddings, and the final linear transformation.

### E.3  SENTENCE EMBEDDING MODELS

For all four sentence encoding models, we used publicly available and freely downloadable pre-trained versions found at the following links:

- InferSent: `https://github.com/facebookresearch/InferSent`
- Skip-thought: `https://github.com/ryankiros/skip-thoughts`
- SST: `https://nlp.stanford.edu/software/corenlp.shtml`

- SPINN: `https://github.com/stanfordnlp/spinn`

InferSent is a bidirectional LSTM with 4096-dimensional hidden states. For Skip-thought, we use the unidirectional variant, which is an LSTM with 2400-dimensional hidden states. The SST model is a recurrent neural tensor network (RNTN) with 25-dimensional hidden states. Finally, for SPINN, we use the SPINN-PI-NT version, which is equivalent to a tree-LSTM (Tai et al., 2015) with 300-dimensional hidden states.

### E.4 TPDNs trained on sentence models

For training a TPDN to approximate the sentence encoding models, the filler embedding dimensions were dictated by the size of the pretrained word embeddings; these dimensions were 300 for InferSent and SPINN, 620 for Skip-thought, and 25 for SST. The linear transformation applied to the word embeddings did not change their size. For role embedding dimensionality we tested all role dimensions in $\{1, 5, 10, 20, 40, 60\}$. The best-performing dimension was chosen based on preliminary experiments and used for all subsequent experiments; we thereby chose role dimensionalities of 10 for InferSent and Skip-thought, 20 for SST, and 5 for SPINN. In general, role embedding dimensionalities of 5, 10, and 20 all performed noticeably better than 1, 40, and 60, but there was not much difference between 5, 10, and 20.

The training regimen for the TPDNs on sentence models was the same as for the TPDNs trained on digit sequences. The TPDNs were given the sequences of fillers (i.e. the words), the roles hypothesized to go with those fillers, the sequence embeddings produced by the RNN, the initial pretrained word embeddings, the dimensionalities of the linearly-transformed filler embeddings, the role embeddings, and the final linear transformation. The parameters that were updated by training were the specific values for the role embeddings, the linear transformation that was applied to the pretrained word embeddings, and the final linear transformation.

The sentences whose encodings we trained the TPDNs to approximate were the premise sentences from the SNLI corpus (Bowman et al., 2015). We also tried instead using the sentences in the WikiText-2 corpus (Merity et al., 2016) but found better performance with the SNLI sentences. This is plausibly because the shorter, simpler sentences in the SNLI corpus made it easier for the model to learn the role embeddings without distraction from the fillers.

## F Downstream task performance for TPDNs approximating sentence encoders

For each TPDN trained to approximate a sentence encoder, we evaluate it on four downstream tasks: (i) Stanford Sentiment Treebank (SST), which is labeling the sentiment of movie reviews (Socher et al., 2013); this task is further subdivided into SST2 (labeling the reviews as *positive* or *negative*) and SST5 (labeling the reviews on a 5-point scale, where 1 means *very negative* and 5 means *very positive*). The metric we report for both tasks is accuracy. (ii) Microsoft Research Paraphrase Corpus (MRPC), which is labeling whether two sentences are paraphrases of each other (Dolan et al., 2004). For this task, we report both accuracy and F1. (iii) Semantic Textual Similarity Benchmark (STS-B), which is giving a pair of sentences a score on a scale from 0 to 5 indicating how similar the two sentences are (Cer et al., 2017). For this task, we report Pearson and Spearman correlation coefficients. (iv) Stanford Natural Language Inference (SNLI), which involves labeling a pair of sentences to indicate whether the first entails the second, contradicts the second, or neither (Bowman et al., 2015). For this task, we report accuracy as the evaluation metric.

### F.1 Substitution performance

The first results we report for the TPDN approximations of sentence encoders is similar to the substitution accuracy used for digit encoders. Here, we use SentEval (Conneau & Kiela, 2018) to train linear classifiers for all downstream tasks on the original sentence encoding model; then, we freeze the weights of these classifiers and use them to classify the test-set encodings generated by the TPDN approximation. We use the classifier parameters recommended by the SentEval authors: using a linear classifier (not a multi-layer perceptron) trained with the Adam algorithm (Kingma &

Ba, 2015) using a batch size of 64, a tenacity of 5, and an epoch size of 4. The results are shown in Table 8.

## F.2 Agreement between the TPDN and the original model

Next, we analyze the same results from the previous section, but instead of reporting accuracies we report the extent to which the TPDN's predictions agree with the original model's predictions. For SST, MRPC, and SNLI, this agreement is defined as the proportion of their labels that are the same. For STS-B, the agreement is the Pearson correlation between the original model's outputs and the TPDN's outputs. The results are in Table 9.

## F.3 Training a classifier on the TPDN

Finally, we consider treating the TPDNs as models in their own right and use SentEval to both train and test downstream task classifiers on the TPDNs. The results are in Table 10.

| Filler dim. | Role dim. | Without linear layer | With linear layer |
|---|---|---|---|
| 1 | 60 | 0.0002 | 0.003 |
| 2 | 30 | 0 | 0.042 |
| 3 | 20 | 0.001 | 0.82 |
| 4 | 15 | 0.0006 | 0.80 |
| 5 | 12 | 0 | 0.90 |
| 6 | 10 | 0 | 0.92 |
| 10 | 6 | 0.0002 | 0.99 |
| 12 | 5 | 0 | 0.67 |
| 15 | 4 | 0.0002 | 0.37 |
| 20 | 3 | 0 | 0.14 |
| 30 | 2 | 0.0002 | 0.02 |
| 60 | 1 | 0.001 | 0.0014 |

Table 4: Substitution accuracies with and without the final linear layer, for TPDNs using various combinations of filler and role embedding dimensionality. These TPDNs were approximating a seq2seq model trained to perform reversal.

| | LTR | RTL | Bi | Wickel | Tree | BOW | Parameters |
|---|---|---|---|---|---|---|---|
| Tensor product (20 dim.) | 0.054 | 0.993 | 0.996 | 0.175 | 0.188 | 0.002 | 24k |
| Tensor product (60 dim.) | 0.046 | 0.988 | 0.996 | 0.138 | 0.172 | 0.001 | 217k |
| Circular convolution (20 dim.) | 0.004 | 0.045 | 0.000 | 0.000 | 0.003 | 0.001 | 1.5k |
| Circular convolution (60 dim.) | 0.048 | 0.964 | 0.066 | 0.001 | 0.013 | 0.001 | 4.6k |
| Elementwise product (20 dim.) | 0.026 | 0.617 | 0.386 | 0.024 | 0.027 | 0.001 | 1.5k |
| Elementwise product (60 dim.) | 0.051 | 0.992 | 0.993 | 0.120 | 0.173 | 0.001 | 4.6k |

Table 5: Approximating a unidirectional seq2seq model trained to perform sequence reversal: substitution accuracies using different binding operations.

| Encoder | Decoder | Autoencode | Reverse | Sort | Interleave |
|---|---|---|---|---|---|
| Uni | Uni | 0.999 | 1.000 | 1.000 | 1.000 |
| Uni | Bi | 0.949 | 0.933 | 1.000 | 0.968 |
| Uni | Tree | 0.979 | 0.967 | 0.970 | 0.964 |
| Bi | Uni | 0.993 | 0.999 | 1.000 | 0.995 |
| Bi | Bi | 0.834 | 0.883 | 1.000 | 0.939 |
| Bi | Tree | 0.967 | 0.920 | 0.959 | 0.909 |
| Tree | Uni | 0.981 | 0.978 | 1.000 | 0.987 |
| Tree | Bi | 0.891 | 0.900 | 1.000 | 0.894 |
| Tree | Tree | 0.989 | 0.962 | 0.999 | 0.934 |

Table 6: Accuracies of the various sequence-to-sequence encoder/decoder combinations at the different training tasks. Each number in this table is an average across five random initializations. Uni = unidirectional; bi = bidirectional.

| Task | Encoder | Decoder | LTR | RTL | Bi | Wickel | Tree | BOW |
|------|---------|---------|-----|-----|-----|--------|------|-----|
| Autoencode | Uni | Uni | 0.871 | 0.112 | **0.992** | 0.279 | 0.246 | 0.001 |
| | Uni | Bi | 0.275 | 0.273 | **0.400** | 0.400 | 0.238 | 0.005 |
| | Uni | Tree | 0.053 | 0.086 | 0.094 | 0.105 | **0.881** | 0.009 |
| | Bi | Uni | 0.748 | 0.136 | **0.921** | 0.209 | 0.207 | 0.002 |
| | Bi | Bi | 0.097 | 0.124 | **0.179** | 0.166 | 0.128 | 0.006 |
| | Bi | Tree | 0.051 | 0.081 | 0.088 | 0.095 | **0.835** | 0.007 |
| | Tree | Uni | 0.708 | 0.330 | **0.865** | 0.595 | 0.529 | 0.007 |
| | Tree | Bi | 0.359 | 0.375 | 0.491 | **0.569** | 0.376 | 0.009 |
| | Tree | Tree | 0.041 | 0.069 | 0.076 | 0.095 | **0.958** | 0.009 |
| Reverse | Uni | Uni | 0.062 | 0.986 | **0.995** | 0.177 | 0.204 | 0.002 |
| | Uni | Bi | 0.262 | 0.268 | 0.386 | **0.406** | 0.228 | 0.006 |
| | Uni | Tree | 0.103 | 0.112 | 0.177 | 0.169 | **0.413** | 0.002 |
| | Bi | Uni | 0.037 | 0.951 | **0.965** | 0.084 | 0.146 | 0.001 |
| | Bi | Bi | 0.121 | 0.140 | **0.228** | 0.170 | 0.140 | 0.005 |
| | Bi | Tree | 0.085 | 0.105 | 0.17 | 0.151 | **0.385** | 0.002 |
| | Tree | Uni | 0.178 | 0.755 | **0.802** | 0.424 | 0.564 | 0.007 |
| | Tree | Bi | 0.302 | 0.332 | 0.442 | **0.549** | 0.368 | 0.009 |
| | Tree | Tree | 0.083 | 0.096 | 0.147 | 0.152 | **0.612** | 0.004 |
| Sort | Uni | Uni | 0.895 | 0.895 | **0.923** | 0.878 | 0.890 | 0.892 |
| | Uni | Bi | 0.898 | 0.894 | **0.923** | 0.916 | 0.915 | 0.904 |
| | Uni | Tree | 0.218 | 0.212 | 0.207 | 0.193 | **0.838** | 0.275 |
| | Bi | Uni | 0.886 | 0.884 | **0.917** | 0.812 | 0.871 | 0.847 |
| | Bi | Bi | 0.921 | 0.925 | **0.945** | 0.835 | 0.927 | 0.934 |
| | Bi | Tree | 0.219 | 0.216 | 0.209 | 0.194 | **0.816** | 0.273 |
| | Tree | Uni | 0.997 | 0.998 | 0.997 | **0.999** | 0.999 | 0.998 |
| | Tree | Bi | **1.000** | 1.000 | 0.997 | 1.000 | 1.000 | 1.000 |
| | Tree | Tree | 0.201 | 0.199 | 0.179 | 0.181 | **0.978** | 0.249 |
| Interleave | Uni | Uni | 0.269 | 0.181 | **0.992** | 0.628 | 0.357 | 0.003 |
| | Uni | Bi | 0.177 | 0.095 | **0.728** | 0.463 | 0.255 | 0.005 |
| | Uni | Tree | 0.040 | 0.033 | 0.116 | 0.089 | **0.373** | 0.003 |
| | Bi | Uni | 0.186 | 0.126 | **0.965** | 0.438 | 0.232 | 0.001 |
| | Bi | Bi | 0.008 | 0.074 | **0.600** | 0.128 | 0.162 | 0.002 |
| | Bi | Tree | 0.031 | 0.025 | 0.069 | 0.057 | **0.395** | 0.004 |
| | Tree | Uni | 0.330 | 0.208 | **0.908** | 0.663 | 0.522 | 0.005 |
| | Tree | Bi | 0.191 | 0.151 | **0.643** | 0.518 | 0.391 | 0.006 |
| | Tree | Tree | 0.027 | 0.025 | 0.059 | 0.069 | **0.606** | 0.004 |

Table 7: Substitution accuracies for TPDNs applied to all combinations of encoder, decoder, training task, and hypothesized role scheme. Each number is an average across five random initializations. Uni = unidirectional; bi = bidirectional.

|  | LTR | RTL | Bi | Tree | BOW | Original |
|---|---|---|---|---|---|---|
| **InferSent** | | | | | | |
| SST2 | 0.79 | 0.79 | 0.77 | 0.79 | 0.80 | 0.85 |
| SST5 | 0.40 | 0.39 | 0.40 | 0.41 | 0.42 | 0.46 |
| MRPC (accuracy) | 0.70 | 0.71 | 0.72 | 0.70 | 0.72 | 0.73 |
| MRPC (F1) | 0.81 | 0.82 | 0.82 | 0.80 | 0.81 | 0.81 |
| STS-B (Pearson) | 0.69 | 0.70 | 0.71 | 0.70 | 0.69 | 0.78 |
| STS-B (Spearman) | 0.68 | 0.68 | 0.69 | 0.68 | 0.67 | 0.78 |
| SNLI | 0.71 | 0.69 | 0.72 | 0.71 | 0.66 | 0.84 |
| **Skip-thought** | | | | | | |
| SST2 | 0.58 | 0.51 | 0.50 | 0.50 | 0.61 | 0.81 |
| SST5 | 0.29 | 0.27 | 0.21 | 0.25 | 0.31 | 0.43 |
| MRPC (accuracy) | 0.60 | 0.62 | 0.62 | 0.61 | 0.66 | 0.74 |
| MRPC (F1) | 0.73 | 0.75 | 0.76 | 0.75 | 0.79 | 0.82 |
| STS-B (Pearson) | -0.01 | -0.07 | -0.10 | -0.05 | 0.06 | 0.73 |
| STS-B (Spearman) | 0.23 | -0.01 | -0.05 | 0.00 | 0.08 | 0.72 |
| SNLI | 0.35 | 0.35 | 0.34 | 0.35 | 0.35 | 0.73 |
| **SST** | | | | | | |
| SST2 | 0.76 | 0.76 | 0.76 | 0.75 | 0.77 | 0.83 |
| SST5 | 0.37 | 0.38 | 0.37 | 0.37 | 0.38 | 0.45 |
| MRPC (accuracy) | 0.67 | 0.67 | 0.67 | 0.66 | 0.66 | 0.66 |
| MRPC (F1) | 0.80 | 0.80 | 0.80 | 0.79 | 0.80 | 0.80 |
| STS-B (Pearson) | 0.24 | 0.21 | 0.22 | 0.19 | 0.24 | 0.29 |
| STS-B (Spearman) | 0.24 | 0.22 | 0.23 | 0.20 | 0.25 | 0.27 |
| SNLI | 0.40 | 0.41 | 0.41 | 0.41 | 0.40 | 0.42 |
| **SPINN** | | | | | | |
| SST2 | 0.73 | 0.73 | 0.73 | 0.74 | 0.74 | 0.76 |
| SST5 | 0.36 | 0.36 | 0.35 | 0.37 | 0.37 | 0.39 |
| MRPC (accuracy) | 0.67 | 0.68 | 0.67 | 0.67 | 0.68 | 0.70 |
| MRPC (F1) | 0.75 | 0.78 | 0.76 | 0.76 | 0.76 | 0.79 |
| STS-B (Pearson) | 0.60 | 0.60 | 0.62 | 0.62 | 0.53 | 0.67 |
| STS-B (Spearman) | 0.58 | 0.59 | 0.59 | 0.59 | 0.57 | 0.65 |
| SNLI | 0.67 | 0.67 | 0.68 | 0.69 | 0.54 | 0.79 |

Table 8: Substitution results on performing the applied tasks for TPDNs trained to approximate the representations from each of the four downloaded models. For MRPC, we report accuracy and F1. For STS-B, we report Pearson correlation and Spearman correlation. All other metrics are accuracies.

|  | LTR | RTL | Bi | Tree | BOW | Original |
|---|---|---|---|---|---|---|
| **Infersent** | | | | | | |
| SST2 | 0.85 | 0.84 | 0.82 | 0.83 | 0.84 | 1.00 |
| SST5 | 0.60 | 0.59 | 0.58 | 0.59 | 0.61 | 1.00 |
| MRPC | 0.78 | 0.80 | 0.78 | 0.77 | 0.79 | 1.00 |
| STS-B | 0.87 | 0.86 | 0.87 | 0.86 | 0.84 | 1.00 |
| SNLI | 0.77 | 0.74 | 0.77 | 0.77 | 0.71 | 1.00 |
| **Skip-thought** | | | | | | |
| SST2 | 0.58 | 0.54 | 0.50 | 0.51 | 0.60 | 1.00 |
| SST5 | 0.41 | 0.41 | 0.18 | 0.35 | 0.43 | 1.00 |
| MRPC | 0.69 | 0.71 | 0.74 | 0.72 | 0.79 | 1.00 |
| STS-B | -0.10 | -0.12 | -0.16 | -0.13 | -0.04 | 1.00 |
| SNLI | 0.37 | 0.37 | 0.36 | 0.36 | 0.37 | 1.00 |
| **SST** | | | | | | |
| SST2 | 0.84 | 0.84 | 0.83 | 0.84 | 0.85 | 1.00 |
| SST5 | 0.65 | 0.64 | 0.64 | 0.64 | 0.65 | 1.00 |
| MRPC | 0.99 | 0.99 | 0.99 | 0.98 | 0.99 | 1.00 |
| STS-B | 0.64 | 0.59 | 0.62 | 0.68 | 0.60 | 1.00 |
| SNLI | 0.48 | 0.51 | 0.49 | 0.67 | 0.49 | 1.00 |
| **SPINN** | | | | | | |
| SST2 | 0.77 | 0.77 | 0.77 | 0.79 | 0.79 | 1.00 |
| SST5 | 0.61 | 0.62 | 0.63 | 0.63 | 0.62 | 1.00 |
| MRPC | 0.68 | 0.73 | 0.72 | 0.74 | 0.70 | 1.00 |
| STS-B | 0.72 | 0.73 | 0.74 | 0.76 | 0.70 | 1.00 |
| SNLI | 0.72 | 0.72 | 0.73 | 0.76 | 0.58 | 1.00 |

Table 9: The proportion of times that a classifier trained on a sentence encoding model gave the same downstream-task predictions based on the original sentence encoding model and based on a TPDN approximating that model, where the TPDN uses the role schemes indicated by the column header. For all tasks but STS-B, these numbers show the proportion of predictions that matched; chance performance is 0.5 for SST2 and MRPC, 0.2 for SST5, and 0.33 for SNLI. For STS-B, the metric shown is the Pearson correlation between the TPDN's similarity ratings and the original model's similarity ratings; chance performance here is 0.0.

|  | LTR | RTL | Bi | Tree | BOW | Original |
|---|---|---|---|---|---|---|
| **Infersent** | | | | | | |
| SST2 | 0.82 | 0.82 | 0.81 | 0.81 | 0.83 | 0.85 |
| SST5 | 0.44 | 0.44 | 0.44 | 0.44 | 0.43 | 0.46 |
| MRPC (acc.) | 0.71 | 0.73 | 0.72 | 0.70 | 0.73 | 0.73 |
| MRPC (F1) | 0.80 | 0.81 | 0.81 | 0.80 | 0.81 | 0.81 |
| STS-B (Pearson) | 0.71 | 0.71 | 0.71 | 0.71 | 0.71 | 0.78 |
| STS-B (Spearman) | 0.69 | 0.70 | 0.70 | 0.69 | 0.70 | 0.78 |
| SNLI | 0.77 | 0.76 | 0.77 | 0.77 | 0.75 | 0.84 |
| **Skip-thought** | | | | | | |
| SST2 | 0.59 | 0.56 | 0.50 | 0.52 | 0.61 | 0.81 |
| SST5 | 0.30 | 0.30 | 0.25 | 0.28 | 0.32 | 0.43 |
| MRPC (acc.) | 0.67 | 0.66 | 0.66 | 0.66 | 0.67 | 0.74 |
| MRPC (F1) | 0.80 | 0.80 | 0.80 | 0.80 | 0.79 | 0.82 |
| STS-B (Pearson) | 0.19 | 0.17 | 0.13 | 0.13 | 0.23 | 0.73 |
| STS-B (Spearman) | 0.17 | 0.16 | 0.08 | 0.10 | 0.20 | 0.72 |
| SNLI | 0.47 | 0.46 | 0.46 | 0.44 | 0.46 | 0.73 |
| **SST** | | | | | | |
| SST2 | 0.78 | 0.78 | 0.77 | 0.78 | 0.79 | 0.83 |
| SST5 | 0.40 | 0.40 | 0.40 | 0.39 | 0.41 | 0.45 |
| MRPC (acc.) | 0.67 | 0.67 | 0.66 | 0.66 | 0.67 | 0.66 |
| MRPC (F1) | 0.80 | 0.80 | 0.80 | 0.80 | 0.80 | 0.88 |
| STS-B (Pearson) | 0.28 | 0.23 | 0.23 | 0.20 | 0.28 | 0.29 |
| STS-B (Spearman) | 0.27 | 0.23 | 0.23 | 0.20 | 0.27 | 0.27 |
| SNLI | 0.45 | 0.44 | 0.45 | 0.43 | 0.45 | 0.42 |
| **SPINN** | | | | | | |
| SST2 | 0.79 | 0.78 | 0.77 | 0.77 | 0.80 | 0.76 |
| SST5 | 0.42 | 0.42 | 0.40 | 0.42 | 0.42 | 0.39 |
| MRPC (acc.) | 0.72 | 0.71 | 0.71 | 0.68 | 0.72 | 0.70 |
| MRPC (F1) | 0.81 | 0.80 | 0.80 | 0.79 | 0.80 | 0.79 |
| STS-B (Pearson) | 0.68 | 0.67 | 0.67 | 0.67 | 0.66 | 0.67 |
| STS-B (Spearman) | 0.67 | 0.66 | 0.66 | 0.65 | 0.65 | 0.65 |
| SNLI | 0.72 | 0.71 | 0.72 | 0.73 | 0.71 | 0.79 |

Table 10: Downstream task performance for classifiers trained and tested on the TPDNs that were trained to approximate each of the four applied models. The rightmost column indicates the performance of the original model (without the TPDN approximation).

