# OpenReview forum: "RNNs implicitly implement tensor-product representations"
_ICLR.cc/2019/Conference_

### Official Review · AnonReviewer3 · 2018-10-31
**An interesting paper in general**

**Rating:** 6
**Confidence:** 4

**Review:**

This paper presents an analysis of popularly-use RNN model for structure modeling abilities by designing Tensor Product Decomposition Networks to approximate the encoder. The results show that the representations exhibit interpretable compositional structure. To provide better understanding, the paper evaluates the performance on synthesized digit sequence data as well as several sentence-encoding tasks.

Pros:
1. The paper is well-written and easy to follow. The design of the TPDN and corresponding settings (including what an filler is and what roles are included) for experiments are understandable. It makes good point at the end of the paper (section 4) on how these analysis contribute to further design of RNN models, which seems useful.
2. The experiments are extensive to support their claims. Not only synthetic data but also several popularly-used data and models are being conducted and compared. An addition of analogy dataset further demonstrate the effect of TPDN on modeling structural regularities.

Cons:
1. More detailed and extensive discussion on the contribution of the paper should be included in the introduction part to help readers understand what's the point of investigating TPDN on RNN models.
2. Some details are missing to better understand the construction. For example, on page 4, Evaluation, it is unclear of how TPDN encoder is trained, specifically, which parameters are updated? What's the objective for training? It is also unclear of whether the models in Figure 3(c) use bidirectional or unidirectional or tree decoder? In Section 3, it could be better to roughly introduce each of the existing 4 models. How do TPDN trained for these 4 sentence encoding models need to be further illustrated. More reasons should be discussed for the results in Table 2 (why bag-of-words role seem to be ok, why skip-thought cannot be approximated well).
3. It could be better to provide the actual performance (accuracy) given by TPDN on the 4 existing tasks.
4. Further thoughts: have you considered applying these analysis on other models besides RNN?

---

> ### Author Response · Authors · 2018-11-11
> **Author response**
>
> Thank you for the feedback! Here are replies to the concerns you raise:
>
> Point 1:
> We will edit the introduction to make the contributions clearer.
>
> Point 2:
> Some of these details are available in the appendices, and we will add the ones that are not already there. We will also make it clearer in the main text that such information is available in the appendices.
>
> We will also clarify our discussion of the results in Table 2. We do not have a strong hypothesis for why Skip-thought is approximated less well than the other models. For the other models, our conjecture is that the models’ representations consist of a combination of a bag-of-words representation and some structural information that is occasionally, but not reliably, present as well. This conjecture is consistent with the finding that these representations could be approximated well, though not perfectly, with a bag-of-words role scheme.
>
> We argue that such representations arise because the training tasks for these sentence embedding models do not depend much on the structure of the input; our results in Table 3b indicate that only structure-sensitive training tasks will induce models to learn structured representations.
>
> However, we will also clarify the other two possible explanation for the results in Table 2, namely that the models could be well-approximated by some role scheme that we did not test, or that the models are using some systematic but non-TPR structural representation.
>
> Point 3:
> Tables 9 and 10 show the actual performance on downstream tasks of TPDNs trained to approximate the sentence embedding models. We did not emphasize these results, however, because we are presenting the TPDN as a tool for analyzing existent models, not as a new architecture for performing tasks of interest. Therefore, the most relevant metrics are ones showing how the TPDN approximates existing models, not how it performs in its own right. For this same reason, we have not tried training the TPDN end-to-end on these specific tasks rather than training it to approximate existing models.
>
> Point 4:
> Yes, we have considered applying the TPDN to other models.
>
> For example, TPDNs might be used to analyze transformer models by seeing whether the representations generated for each word via self-attention can be approximated as tensor product representations based on the structure of the surrounding context. We are further interested in expanding the domain of inquiry to computer vision to see if convolutional neural networks learn structured representations of scenes that can be approximated by tensor product representations.
>
> Finally, we hope that, by making our code available on GitHub, we will enable others to use this technique to analyze the models they are interested in.

---

> > ### Author Response · Authors · 2018-11-27
> > **Revisions uploaded**
> >
> > Thank you again for the suggestions. We have uploaded a new version of the paper that incorporates the changes discussed in our response.

---

### Official Review · AnonReviewer2 · 2018-11-03
**An interesting work offers first step in inspecting RNN representations, the experimental results does not fully support the claim**

**Rating:** 6
**Confidence:** 4

**Review:**

The work proposes Tensor Product Decomposition Networks (TRDN) as a way to uncover the representation learned in recurrent neural networks (RNNs). TRDN trains a Tensor Product Representation, which additively combine tensor products of role (e.g., sequence position) embeddings and filler (e.g., word) embeddings to approximate the encoding produced by RNNs. TRDN as a result shed light into inspecting and interpreting representation learned through RNNs. The authors suggest that the structures captured in RNNs are largely compositional and can be well captured by TPRs without recurrence and nonlinearity.

pros:
1. The paper is mostly clearly written and easy to follow. The diagrams shown in Figure 2 are illustrative;
2. TRDN offers a headway to look into and interpret the representations learned in RNNs, which remained largely incomprehensible;
3. The analysis and insight provided in section 4 is interesting and insightful. In particular, how does the training task influence the kinds of structural representation learned.


cons:
1. The method relies heavily on these manually crafted role schemes as shown in section 2.1; It is unclear the gap in the approximation of TPRs to the encodings learned in RNNs are due to inaccurate role definition or in fact RNNs learn more complex structural dependencies which TPRs cannot capture;
2. The MSE of approximation error shown in Table 1 are not informative. How should these numbers be interpreted? Why normalizing by dividing by the MSE from training TPDN on random vectors?
3. The alignment between prediction using RNN representations and TPDN approximations shown in Table 2 are far from perfect, which would contradict with the claim that RNNs only learn tensor-product representation.

---

> ### Author Response · Authors · 2018-11-11
> **Author response**
>
> Thank you for these comments! Here are replies to the specific concerns you discuss:
>
> Point 1:
> There are two issues raised here. The first is the limitation of using handcrafted role schemes. What this paper attempts to do is explicit, discrete model comparisons between different candidate role schemes. We take this to be a necessary first step on the way to automatically exploring the space of logically possible role schemes, and thus "learning" the optimal role scheme, thereby ruling out this kind of omission.
>
> However, such a project is an ambitious goal, and we feel it is important to establish the basic methodology, and some basic results, first. Figure 3c and Table 3b show cases where handcrafted role schemes have succeeded near-perfectly, serving as a proof of concept that, given the right role scheme (whether it be hand-crafted or learned), TPDNs can reveal striking levels of systematic structure in RNN representations.
>
> The second issue is the possibility that RNNs do use a systematic structural representation whose representational space cannot be approximated with a TPR. We agree that this is a possibility; although TPRs are capable of capturing complex structural relations, they rely upon certain assumptions about the structure of the representational space. RNNs are not constrained in any way that enforces these assumptions - indeed, this fact is partly why we find the successful TPDN approximations so striking in Figure 3c and Table 3b.
>
> In the final version of the paper, we will emphasize the possibility that RNNs may sometimes use non-TPR structural representations.
>
> Point 2:
> The MSE is informative on a relative level: It allows us to compare role schemes within a model. To allow comparisons across models, we normalize by dividing by the random-vector performance to factor out overall vector magnitude differences across different models. The other metrics besides MSE allow for absolute measurements of performance. We will clarify the contributions of these different metrics.
>
> Point 3:
> We will edit the paper to clarify the three possibilities for why the alignments in Table 2 are not perfect.
>
> Two of the possibilities, as discussed in our response to your first point, are that the RNNs are using some role scheme other than the ones we tested, or that the RNNs are using some structural representation that cannot be approximated with any tensor product representation.
>
> However, we argue for a third possibility: the representation can be characterized as a combination of a bag-of-words representation, plus some incomplete (not always encoded) structural information. Such a result is consistent with our observation that bag-of-words roles yield a strong but imperfect approximation for the sentence embedding models. We will edit the text to emphasize that this is merely a conjecture and that the other two possibilities must also be considered.
>
> Finally, we agree with your comment that these results do not indicate that RNNs *only* learn tensor-product representations, but we had not intended to make that claim (we meant the title to be read as “RNNs *sometimes* implement tensor-product representations”).

---

> > ### Author Response · Authors · 2018-11-27
> > **Revisions uploaded**
> >
> > Thank you again for your comments. We have uploaded a new version of the paper that incorporates the changes discussed in our response.

---

### Official Review · AnonReviewer1 · 2018-11-04
**RNNs implicitly implement tensor-product representations**

**Rating:** 7
**Confidence:** 4

**Review:**

This paper is not standalone.  A section on the basics of document analysis would have been nice.

---

> ### Author Response · Authors · 2018-11-10
> **Author response**
>
> Thank you for the feedback. We believe it would be difficult to make a paper completely stand-alone, but it is, indeed, not our goal to discuss sentence/sequence embeddings per se (note that the models we use are sentence models, not document models), but, rather, to describe a general analysis method applied to the special case of these models.
>
> To help make the paper understandable with less context, we will integrate a very short description of what we currently refer to as "the standard left-to-right sequence-to-sequence setup" on page 3.

---

> > ### Author Response · Authors · 2018-11-27
> > **Revisions uploaded**
> >
> > We have uploaded a revised version of the paper that incorporates the change mentioned above.

---

### Author Response · Authors · 2018-12-08
**Interactive demo uploaded**

We have created an anonymized webpage with interactive demos to accompany this paper. The page can be found here:
https://tpdn-iclr.github.io/tpdn-demo/tpr_demo.html

---

### Meta-Review · Area_Chair1 · 2018-12-15
**Interesting work however it lacks connection with modern tensor models.**

**Confidence:** 5
**Recommendation:** Accept (Poster)

**Metareview:**

AR1 seeks the paper to be more standalone and easier to read. As this comment comes from the reviewer who is very experienced in tensor models, it is highly recommended that the authors make further efforts to make the paper easier to follow. AR2 is concerned about  the manually crafted role schemes and alignment discrepancy of results between these schemes and RNNs. To this end, the authors hypothesized further reasons as to why this discrepancy occurs. AC encourages authors to make further efforts to clarify this point without overstating the ability of tensors to model RNNs (it would be interesting to see where these schemes and RNN differ). Lastly, AR3 seeks more clarifications on contributions.

While the paper is not ground breaking, it offers some starting point on relating tensors and RNNs. Thus, AC recommends an accept. Kindly note that tensor outer products have been used heavily in computer vision, i.e.:
- Higher-Order Occurrence Pooling for Bags-of-Words: Visual Concept Detection by Koniusz et al. (e.g. section 3 considers bi-modal outer tensor product for combining multiple sources: one source can be considered a filter, another as role (similar to Smolensky at al. 1990), e.g. a spatial grid number refining local role of a visual word. This further is extended to multi-modal cases (multiple filter or role modes etc.) )
- Multilinear image analysis for facial recognition (e.g. so called tensor-faces) by Vasilescu et al.
- Multilinear independent components analysis by Vasilescu et al.
- Tensor decompositions for learning latent variable models by Anandkumar et al.

Kindly  make connections to these works in your final draft (and to more prior works).